# Radial Basis Feature Transformation to Arm CNNs Against Adversarial Attacks

## Abstract

The linear and non-flexible nature of deep convolutional models makes them vulnerable to carefully crafted adversarial perturbations. To tackle this problem, in this paper, we propose a nonlinear radial basis convolutional feature transformation by learning the Mahalanobis distance function that maps the input convolutional features from the same class into tight clusters. In such a space, the clusters become compact and well-separated, which prevent small adversarial perturbations from forcing a sample to cross the decision boundary. We test the proposed method on three publicly available image classification and segmentation data-sets namely, MNIST, ISBI ISIC skin lesion, and NIH ChestX-ray14. We evaluate the robustness of our method to different gradient (targeted and untargeted) and non-gradient based attacks and compare it to several non-gradient masking defense strategies. Our results demonstrate that the proposed method can boost the performance of deep convolutional neural networks against adversarial perturbations without accuracy drop on clean data.

## 1 Introduction

It has been shown that deep convolutional neural networks (CNNs) can be highly vulnerable to adversarial perturbations Szegedy et al. (2013); Goodfellow et al. (2015) produced by two main groups of attack mechanisms: white- and back-box, which refer to having access to the victim model parameters and architecture or not, respectively. In order to mitigate the effect of adversarial attacks, two main categories of defense techniques have been proposed: data-level and algorithmic-level. Data-level methods include adversarial training Szegedy et al. (2013); Goodfellow et al. (2015), pre-/post-processing methods (e.g. feature squeezing) Xu et al. (2017), pre-processing using basis functions Shaham et al. (2018), and noise removal Hendrycks & Gimpel (2016); Meng & Chen (2017). Algorithmic-level methods Kolter & Wong (2017); Samangouei et al. (2018); Folz et al. (2018); Samangouei et al. (2018) modify the deep model or training procedure by reducing the magnitude of gradients Papernot et al. (2015) or blocking/masking gradients Buckman et al. (2018); Guo et al. (2017); Song et al. (2017). However, these approaches are not completely effective against several different white- and black-box attacks Samangouei et al. (2018); Tramèr et al. (2017); Meng & Chen (2017) and pre-processing based methods might deteriorate an un-attacked model performance. Generally, most of these defense strategies cause a drop in standard accuracy on clean data Tsipras et al. (2018). For more details on adversarial attacks and defenses, we refer readers to Yuan et al. (2018).

Gradient masking has shown sub-optimal performance against different types of adversarial attacks Papernot et al. (2017); Tramèr et al. (2017). Athalye et al. (2018) identified *obfuscated gradients*, a special case of gradient masking that leads to a false sense of security in defenses against adversarial perturbations. They showed that 7 out of 9 recent white-box defenses relying on this phenomenon (Buckman et al. (2018); Ma et al. (2018); Guo et al. (2017); Dhillon et al. (2018); Xie et al. (2017a); Song et al. (2017); Samangouei et al. (2018)) are vulnerable to single step or non-gradient based attacks. They finally suggested several symptoms of defenses that rely on obfuscated gradients.

As explored in the literature, adversarial examples can be mainly the results of models being *too linear* Goodfellow et al. (2015) in high dimensional manifolds when the decision boundary is close to the manifold of the training data Tanay & Griffin (2016) and/or because of the models' *low flexi-*

*bility* Fawzi et al. (2015). To boost the non-linearity of a model, Goodfellow et al. (2015) explored a variety of methods involving utilizing quadric units and including shallow and deep radial basis function (RBF) networks. They achieved reasonably good performance against adversarial perturbations with shallow RBF networks, however, they found it difficult to train deep RBF models, leading to a high training error using stochastic gradient decent. Fawzi et al. (2018) showed that support vector machines with RBF kernels can effectively resist adversarial attacks.

Typically, a single RBF network layer takes a vector of $\mathbf{x} \in \mathbb{R}^n$ as input and output a scalar function of the input vector $f(\mathbf{x}) : \mathbb{R}^n \to \mathbb{R}$ computed as $f(\mathbf{x}) = \sum_{i=1}^{N} w_i e^{-\beta_i D(\mathbf{x}, \mathbf{c}_i)}$, where $N$ is the number of neurons in the hidden layer, $\mathbf{c}_i$ is the center vector for neuron $i$, $D(\mathbf{x}, \mathbf{c}_i)$ measures the distance between $\mathbf{x}$ and $\mathbf{c}_i$, and $w_i$ weights the output of neuron $i$. The Gaussian basis functions, commonly used in RBF networks, are local to the center vectors, i.e. $\lim_{\|x\| \to \infty} D(\mathbf{x}, \mathbf{c}_i) = 0$, which in turn implies that changing the parameters of a neuron has an increasing smaller effect on the output as the sample input $x$ is farther from center of that neuron (i.e. $x$ is a sample that the RBF network does not 'understand' Goodfellow et al. (2015)). When such a distant sample is in the response domain of an RBF, the change in the response caused by a perturbation $\epsilon$ added to the sample is negligible. Traditional RBF networks are normally trained in two sequential steps. First, an unsupervised method, e.g. k-means clustering, is applied to find RBF centers Wu et al. (2012) and, second, a linear model with coefficients $w_i$ is fit to the desired outputs.

To tackle the *linearity* issue of the current models (i.e. stacking several linear units), particularly deep CNNs, which results in vulnerability to adversarial perturbations, we arm CNNs with radial basis feature transformation. Further, to add more *flexibility* to the models, radial basis functions with Euclidean distance might not be effective as the activation of each neuron depends on the Euclidean distance between a pattern and the neuron center. However, since the activation function is constrained to be symmetrical, all attributes are considered equally relevant. This limitation can be addressed by applying non-symmetrical quadratic distance functions, such as a Mahalanobis distance, in the activation function in order to take into account the variability of the attributes and their correlations. However, computed this distance directly from the variance-covariance matrix of training data is sensitive to outliers and does not consider the accuracy of the learning algorithm.

In this paper, we propose a non-linear radial basis convolutional feature transformation method based on Mahalanobis distance. In contrast to traditional Mahalanobis formulation, in which a constant, pre-defined covariance matrix is adopted, we propose to *learn* such "transformation" matrix $T \in \mathbb{R}^{n \times n}$. To enforce $T$'s positive semi-definiteness, using the eigenvalue decomposition, it can be decomposed into $T'T$. All other RBF parameters, i.e. centers $C = \{c_1, c_2, \cdots, c_N\}$ and $\beta_i$ (width of Gaussian), along with all CNN parameters $\Omega$, are also learned end-to-end using back-propagation. This approach causes the local RBF centers to be adjusted optimally as they are updated based on the whole network parameters. Therefore, we define a loss function $\mathcal{L}$ encoding the classification error in the transformed space and seek $T^*$, $\beta^*$, $C^*$, and $\Omega^*$ that minimize $\mathcal{L}$:

$$T^*, C^*, \beta^*, \Omega^* = \mathrm{argmin}_{P,C,\beta,\Omega} \mathcal{L}(P, C, \beta, \Omega). \tag{1}$$

## 2 RADIAL BASIS FEATURE TRANSFORMATION

Given a convolutional feature map $F$ of size $n \times m \times k$ the goal is to map the features onto a new space $G$ of size $n \times m \times o$ where classes are dense (tightly clustered) and well separated. To achieve this, we leverage a RBF projector that takes as input feature vectors of $F_{i,j,k}$ and transforms them to a new space by learning a transformation matrix under Mahalanobis distance formulation as follows:

$$\phi_k = e^{-\beta D(F_{i,j,K}, c_k)} \tag{2}$$

where $i = 1 : n$, $j = 1 : m$, $\phi_k$ is $k^{th}$ activation function, $c_k$ is $k^{th}$ learnable center, trained $\beta$ controls the width of the Gaussian function and $D(.)$ refers to Mahalanobis distance between a convolutional feature vector and a center computed as:

$$D(F_{i,j,K}, c_k) = \sqrt{(F_{i,j,K} - c_k)^T P (F_{i,j,K} - c_k)} \tag{3}$$

where $P$ refers to learnable transformation matrix of size $n \times m$ and its size varies for different size inputs in each layer. Finally, the projected feature vector $G(F_{i,j,k})$ is computed as:

$$G\left(F_{i,j,K}\right) = \sum_{i=1}^{k} (w_k \cdot \phi_k) + b_k \tag{4}$$

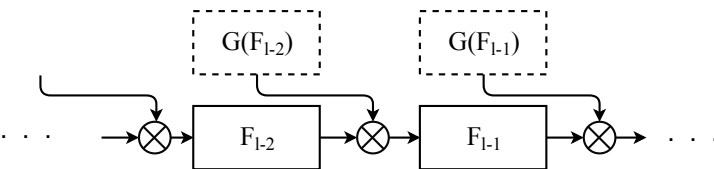

Figure 1: Placing radial basis transformation blocks in a CNN network. $\otimes$ shows concatenation operation. $F_l$ is the output of convolution layer $l$.

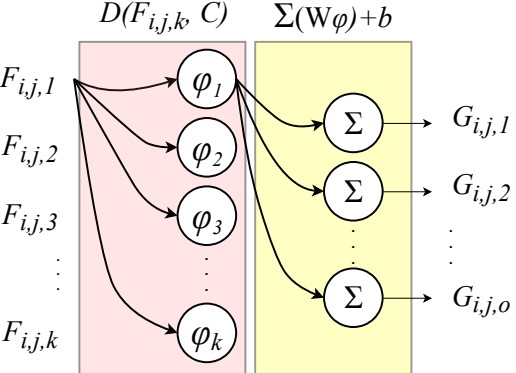

Figure 2: Detailed diagram of the proposed radial basis transformation block $G(F_l)$. In the figure, $i = 1 : n$, $j = 1 : m$ where $n \times m \times k(o)$ is the width, height, and the number of channels of the input (output) feature map.

The detailed diagram of the transformation step is shown in Figure 2. Finally, in each layer $l$ of network, output feature maps of each convolutional block are concatenated with projected feature maps (Figure 1).

## 3 DATA

We conduct two sets of experiments: image (i) classification and (ii) segmentation. (i) For the image classification experiments, we use MNIST and ChestX-ray14 dataset (Wang et al. (2017)), which comprises 112,120 gray-scale images with 14 disease labels and one 'no (clinical) finding' label. We treat all the disease classes as positive and formulate a binary classification task. We randomly selected 90,000 images for training, 45,000 images with "positive" label and the remaining 45,000 with "negative" label. The validation set comprised 13,332 images with 6,666 images of each label. We randomly picked 200 unseen images as the test set, with 93 images with positive and 107 having negative labels. These clean (test) images are used for carrying out different adversarial attacks and the models trained on clean images are evaluated against them. (ii) For the image segmentation task experiments, we used the 2D RGB skin lesion dataset from the 2017 IEEE ISBI International Skin Imaging Collaboration (ISIC) Challenge Codella et al. (2017). We trained on a set of 2,000 images and tested on an unseen set of 150 images.

# 4 EXPERIMENTS AND RESULTS

In this section, we report the results of several experiments for two tasks of classification and segmentation. We first start with MNIST as it has commonly been used for evaluating adversarial attacks and defences. Next we show how the proposed method is scalable to another classification dataset and segmentation task.

## 4.1 EVALUATION ON CLASSIFICATION TASK

In this section, we analyze the proposed method on two different classifications datasets MNIST and of X-chest14. In Table 1, we report the results of the proposed method on MNIST dataset when attacked by different targeted and un-targeted attacks i.e. fast gradient sign method (FGSM) Kurakin et al. (2016a), basic iterative method (BIM) Kurakin et al. (2016b), projected gradient descent (PGD) Madry et al. (2017), Carlini & Wagner method (C&W) Carlini & Wagner (2017), momentum iterative method (MIM) Dong et al. (2018) winner of NIPS 2017 adversarial attacks competition. The proposed method (i.e. PROP) successfully resists all the attacks for which the 3-layers CNN (i.e. ORIG) network almost completely fails e.g. for the strongest attack (i.e. MIM) the proposed method achieves $64.25\%$ accuracy while the original CNN network obtains almost zero ($0.58\%$) accuracy. Further, we test the proposed method with a non-gradient based attack i.e. Gaussian additive noise (GN) Rauber et al. (2017) to show that the robustness of the method is not because of gradient masking.

Table 1: Classification accuracy of different attacks tested on MNIST dataset. FGSM: $\epsilon = 0.3$; BIM: $\epsilon = 0.3$ and iterations = 5; MIM:$\epsilon = 0.3$, iterations = 10, and decay factor = 1; PGD: $\epsilon = 0.1$, iterations=40; C&W: iterations = 50, GM: $\epsilon = 20$.

|      | Clean  | FGSM   | BIM    | MIM    | PGD    | C&W    | GN     |
|------|--------|--------|--------|--------|--------|--------|--------|
| ORIG | 0.9930 | 0.1380 | 0.0070 | 0.0051 | 0.1365 | 0.1808 | 0.7227 |
| PROP | **0.9935** | **0.8582** | **0.7887** | **0.6425** | **0.8157** | **0.9879** | **0.7506** |

To ensure that the proposed method robustness is not due to masked/obfuscated gradient, as suggested by Athalye et al. (2018), we test the proposed feature transformation method based on several characteristic behaviors of defenses which cause obfuscated gradients to occur. a) As reported in Table 1, one-step attacks (e.g. FGSM) did not perform better than iterative attacks (e.g. BIM, MIM); b) According to Tables 4 and 5, black-box attacks did not perform better than white-box ones; c) as shown in Figure 3 (left and middle), larger distortion factors monotonically increase the attack success rate. The right plot in Figure 3 also indicates that the robustness of the proposed method is not because of numerical instability of gradients.

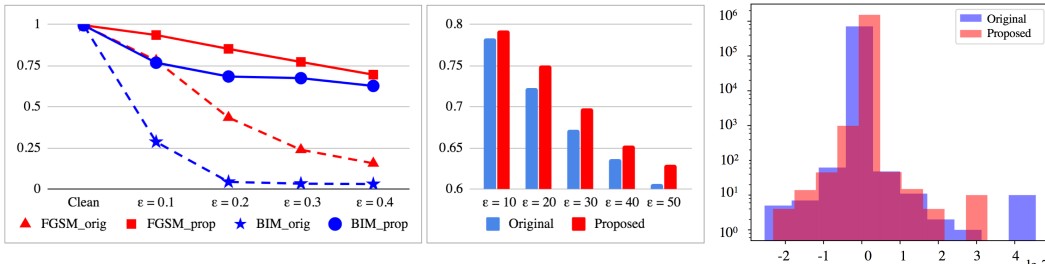

Figure 3: Left: Increasing distortion rate of FGSM and BIM; Middle: Gaussian additive/iterative noise attack with different $\epsilon$ values; Right: Gradient distribution.

Next, to quantify the compactness and separability of different clusters/classes, we evaluate the features produced ORIG and PROP methods with clustering evaluation techniques such as mutual information based score Vinh et al. (2010), homogeneity and completeness Rosenberg & Hirschberg (2007), Silhouette coefficient Rousseeuw (1987), and Calinski-Harabaz index Caliński & Harabasz (1974).

Both Silhouette coefficient and Calinski-Harabaz index quantify how well clusters are separated from each other and how compact they are without taking into account the ground truth labels, while mutual information based score, homogeneity, and completeness scores evaluate clusters based on labels. As reported in Table 2, when the original CNN network applies radial basis feature transformation it achieves considerably higher scores (the higher values for all the metrics the better). As both original and the proposed method achieved high classification test accuracy i.e. $\sim 99\%$, the difference in scores for ground truth label based metrics, i.e. mutual information based, homogeneity, and completeness, scores are small.

Table 2: Feature transformation analysis of MNIST dataset for original 3-layer CNN vs. the proposed method

|  | Silhouette | Calinski | Mutual information | Homogeneity | Completeness |
|---|---|---|---|---|---|
| ORIG | 0.2612 | 1658.20 | 0.9695 | 0.9696 | 0.9721 |
| PROP | **0.4284** | **2570.42** | **0.9720** | **0.9721** | **0.9815** |

In Figure 4, we illustrate feature spaces of each layer in a simple 3-layer CNN network using t-SNE and PCA methods by reducing the high dimensional feature space into two dimensions. As can be seen, the proposed radial basis feature transformation helps reduce intra-class and increase inter-class distances.

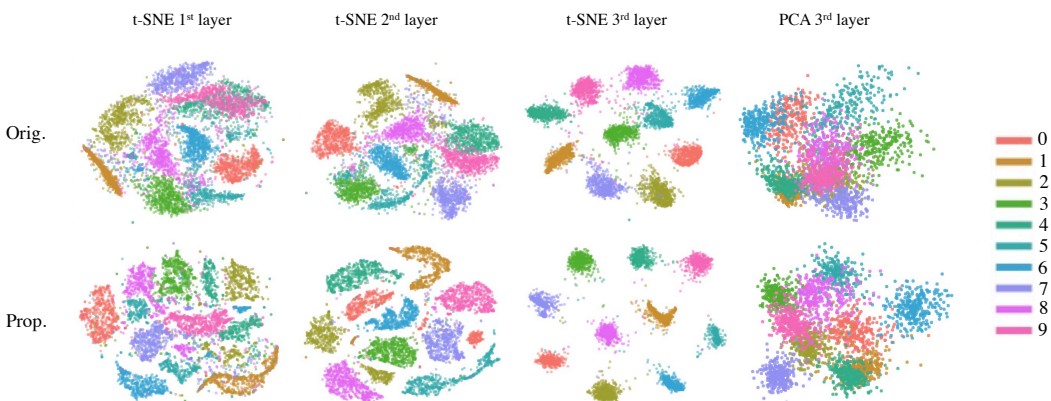

Figure 4: Feature space visualization of MNIST dataset produced via original 3-layer CNN network and the proposed method.

To test the robustness of the proposed method on the X-chest14 dataset we follow the strategy of Taghanaki et al. (2018). We select Inception-ResNet-v2 and modify it by the proposed radial basis function blocks. According to the study done by Taghanaki et al. (2018), we focus on the most effective attacks in term of imperceptibility and power i.e. gradient-based attacks (basic iterative method Kurakin et al. (2016a): BIM and L1-BIM). We also compare the proposed method with two defense strategies: Gaussian data augmentation (GDA) Zantedeschi et al. (2017) and feature squeezing (FSM) Xu et al. (2017). GDA is a data augmentation technique that augments a dataset with copies of the original samples to which Gaussian noise has been added. FSM method reduces the precision of the components of the original input by encoding them on a smaller number of bits. In Table 3, we report the classification accuracy of different attacks and defences (including PROP) on X-chest14 dataset.

## 4.2 EVALUATION ON SEGMENTATION TASK

To assess segmentation vulnerability to adversarial attacks, we apply the recently introduced dense adversary generation (DAG) method proposed by Xie et al. (2017b) to two state-of-the-art segmentation networks i.e. U-Net Ronneberger et al. (2015) and V-Net Milletari et al. (2016) under both white- and black-box conditions. We compare the proposed feature transformation method to other defence strategies e.g. Gaussian and median feature squeezing Xu et al. (2017) (FSG and FSM,

Table 3: Classification accuracy on X-Chest14 dataset for different attacks and defenses

| | | Defence | | | |
|---|---|---|---|---|---|
| Attack | Iteration | ORIG | GDA | FSM | PROP |
| L1-BIM | 5 | 0 | 0 | 0.55 | **0.63** |
| BIM | 5 | 0 | 0 | 0.54 | **0.65** |
| Clean | - | 0.74 | **0.75** | 0.57 | 0.74 |

respectively) and adversarial training Goodfellow et al. (2015) (ADVT). As reported in Table 4, the percentage accuracy (Dice score) drop for the proposed radial basis transformation method is only $1.60\%$ and $6.39\%$ after 10 and 30 iterations of the attack with $\gamma = 0.03$ for U-Net and $8.50\%$ and $13.95\%$ for V-Net, respectively. Applying feature transformation, improved the segmentation accuracy on clean (non-attacked) images from 0.7743 to 0.7780 and 0.8070 to 0.8213 for U-Net and V-Net, respectively.

Table 4: Segmentation results (average DICE±STDV) of different defense mechanisms compared to the proposed radial basis feature transformation method for V-Net and U-Net under DAG attack.

| Network | Method | Clean | 10$i$ (% acc. drop) | 30$i$ (% acc .drop) |
|---|---|---|---|---|
| U-Net | ORIG | $0.7743 \pm 0.2472$ | $0.5594 \pm 0.2405(27.75\%)$ | $0.4396 \pm 0.2715(43.23\%)$ |
| | FSG | $0.7292 \pm 0.2813$ | $0.6382 \pm 0.2524(15.58\%)$ | $0.5858 \pm 0.2668(24.34\%)$ |
| | FSM | $0.7695 \pm 0.2423$ | $0.6039 \pm 0.2436(22.01\%)$ | $0.5396 \pm 0.2587(30.31\%)$ |
| | ADVT | $0.6703 \pm 0.3340$ | $0.7012 \pm 0.3125(9.44\%)$ | $0.6700 \pm 0.3188(13.47\%)$ |
| | PROP | $\mathbf{0.7780 \pm 0.2570}$ | $\mathbf{0.7619 \pm 0.2543(1.60\%)}$ | $\mathbf{0.7248 \pm 0.2767(6.39\%)}$ |
| V-Net | ORIG | $0.8070 \pm 0.2317$ | $0.5320 \pm 0.2535(34.10\%)$ | $0.3865 \pm 0.2663(52.10\%)$ |
| | FSG | $0.7886 \pm 0.2512$ | $0.6990 \pm 0.2324(13.38\%)$ | $0.6840 \pm 0.2298(15.24\%)$ |
| | FSM | $0.8084 \pm 0.2309$ | $0.5928 \pm 0.2556(26.54\%)$ | $0.5144 \pm 0.2675(36.26\%)$ |
| | ADVT | $0.7924 \pm 0.1982$ | $0.7121 \pm 0.2134(11.76\%)$ | $\mathbf{0.7113 \pm 0.2201(11.85\%)}$ |
| | PROP | $\mathbf{0.8213 \pm 0.2164}$ | $\mathbf{0.7384 \pm 0.2078(8.50\%)}$ | $0.6944 \pm 0.2186(13.95\%)$ |

As reported in Table 5, under black-box attack, the proposed method is the best performing method across all 12 experiments except for one in which the accuracy of the best method was just 0.0022 higher (i.e. $0.7284 \pm 0.2682$ vs $0.7262 \pm 0.2621$), however note that the standard deviation of the winner is larger than the proposed method.

Table 5: Segmentation DICE± STDV scores of black-box attacks; adversarial images were produced with methods in first left column and tested with methods in the first row.

| - | U-Net | U-PROP | V-Net | V-PROP |
|---|---|---|---|---|
| U-Net | - | $\mathbf{0.7341 \pm 0.2516}$ | $0.6364 \pm 0.2327$ | $0.7210 \pm 0.2320$ |
| U-PROP | $0.7284 \pm 0.2682$ | - | $0.6590 \pm 0.2676$ | $0.7262 \pm 0.2621$ |
| V-Net | $0.7649 \pm 0.2056$ | $\mathbf{0.7773 \pm 0.2047}$ | - | $0.7478 \pm 0.2090$ |
| V-PROP | $0.7922 \pm 0.2298$ | $\mathbf{0.7964 \pm 0.2353}$ | $0.6948 \pm 0.2163$ | - |

Next, we analyze the usefulness of learning the transformation matrix $T$ and width of the Gaussian ($\beta$) in Mahalanobis distance calculation. As can be seen in Table 6, in all the cases, i.e. testing with clean and 10 and 30 iterations of attack, our method with learned transformation matrix and $\beta$ archived higher performance.

Table 6: Ablation study over the usefulness of learning the transformation matrix $T$ and $\beta$ on the skin lesion dataset for V-Net.

| | $T$ | $\beta$ | Dice±STDV | FPR±STDV | FNR±STDV |
|---|---|---|---|---|---|
| | ✗ | ✓ | $0.7721 \pm 0.2582$ | $0.0149 \pm 0.0278$ | $0.2041 \pm 0.3002$ |
| | ✓ | ✗ | $0.8200 \pm 0.2000$ | $0.0177 \pm 0.0315$ | $0.1547 \pm 0.2297$ |
| Clean | ✗ | ✗ | $0.8002 \pm 0.2248$ | $0.0137 \pm 0.0270$ | $0.1883 \pm 0.2581$ |
| | ✓ | ✓ | $\mathbf{0.8213 \pm 0.2164}$ | $0.0141 \pm 0.0243$ | $0.1706 \pm 0.2450$ |
| | ✗ | ✓ | $0.6471 \pm 0.2603$ | $0.0437 \pm 0.0631$ | $0.1992 \pm 0.3191$ |
| | ✓ | ✗ | $0.7010 \pm 0.1972$ | $0.0606 \pm 0.0656$ | $0.1020 \pm 0.2037$ |
| $10i$ | ✗ | ✗ | $0.6740 \pm 0.2290$ | $0.0458 \pm 0.0453$ | $0.1472 \pm 0.2646$ |
| | ✓ | ✓ | $\mathbf{0.7384 \pm 0.2078}$ | $0.0444 \pm 0.0506$ | $0.1234 \pm 0.2280$ |
| | ✗ | ✓ | $0.6010 \pm 0.2705$ | $0.0371 \pm 0.0360$ | $0.2304 \pm 0.3348$ |
| | ✓ | ✗ | $0.6458 \pm 0.2206$ | $0.0633 \pm 0.0523$ | $0.1164 \pm 0.2247$ |
| $30i$ | ✗ | ✗ | $0.6188 \pm 0.2304$ | $0.0615 \pm 0.0484$ | $0.1384 \pm 0.2515$ |
| | ✓ | ✓ | $\mathbf{0.6944 \pm 0.2186}$ | $0.0418 \pm 0.0463$ | $0.1489 \pm 0.2510$ |

## 5 IMPLEMENTATION DETAILS

**Segmentation experiments.** For both U-Net and V-Net we used batch size of 16, Adadelta optimizer with learning rate of 1.0, rho=0.95, and decay=0.0. We tested DAG method with 10 and 30 iterations and perturbation factor $\gamma = 0.03$. For FSM and FSG defences we applied window size of $3 \times 3$ and standard deviation of 1.0, respectively.

**Classification experiments.** MNIST experiments: for both the original 3-layers CNN (i.e. ORIG) and the proposed method (i.e. PROP = ORIG + feature transformation) we used batch size of 128 and Adam optimizer with learning rate of 0.001. X-chest14 experiments: Inception-ResNet-v2 network was trained from scratch having a batch size of 4 with RMSProp optimizer Toshev & Szegedy (2014) with a decay of 0.9 and $\epsilon = 1$ and an initial learning rate of 0.045, decayed every 2 epochs using an exponential rate of 0.94. For all the gradient based attacks applied in the classification part, we used CleverHans library Nicolas et al. (2018) and for the Gaussian additive noise attack we used FoolBox Rauber et al. (2017).

## 6 CONCLUSION

We proposed a nonlinear radial basis feature transformation method to map convolutional features of each layer in a network into a new space, in which it becomes harder for an attack to find effective gradient directions to perturb the input to fool a model. We evaluated the model under white- and black-box attacks for two different tasks of image classification and segmentation and compared our method to other non-gradient based defenses. We also performed several tests to ensure that the robustness of the proposed method is neither because of numerical instability of gradients nor because of gradient masking. In contrast to previous methods, our proposed feature mapping improved the classification and segmentation accuracy on both clean and perturbed

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
