# OpenReview forum: "Radial Basis Feature Transformation to Arm CNNs Against Adversarial Attacks"
_ICLR.cc/2019/Conference_

### Official Review · AnonReviewer2 · 2018-10-29
**We think that this paper is not of sufficient quality to be accepted in ICLR, for at least the following reasons.**

**Rating:** 3
**Confidence:** 4

**Review:**

The proposed method is too simplistic, the model being succinctly described in less than one page with many errors in the given math expressions. Only the model is given. The optimization problem, as given in (1) is not explained. the authors need to stud the optimization problem, to derive its resolution, and to describe the obtained algorithm.

The authors’ main motivation is to “maps the input convolutional features from the same class into tight clusters. In such a space, the clusters become compact and well-separated, …”. However, in the proposed method is operating in this way. The model is a simple transformation, and nothing ensures the compactness of the feature space, neither the separability of the classes.

It is difficult to understand the “arm CNNs with radial basis feature transformation”. There are two figure in the paper that seek to show this modification of CNN, but this is not enough because nothing is said in the text, which makes these images difficult to understand. Moreover, the figures have notations different than those in the  main body, such as F_{l-1} as opposed to F_{i,j,K}.

What is the transformation to be learned ? Is it T as given in the text before (1), or P as given in (3). In (1), it seems that it is a mix of both, namely T* = argmin_P ! Moreover, it is written “To enforce T’s positive semi-definiteness, using the eigenvalue decomposition, it can be decomposed into T ′T”.  Decomposing T as T’T, means that T is very very special.

Equation (4) is not correct. The summation is on i, which is not in the expressions, but in the result with F_{i,j,K}.

With the exception of Tables 3 and 4, most experiments are on comparing the conventional versus the proposed method. The authors need to compare to other methods available in the literature on defense against adversarial attacks. Moreover, it is not clear why the author compare the proposed method to ADVT (adversarial training) in Table 4, and not in Table 3.

Some references are incomplete. For example, the second reference is missing the publication type, volume, …

---

### Official Review · AnonReviewer3 · 2018-11-02
**Not well written**

**Rating:** 4
**Confidence:** 3

**Review:**

This paper realizes the radial basis function in deep CNNs by leveraging the Mahalanobis distance between a convolutional feature vector and the corresponding center. The method is implemented for the image classification and segmentation tasks against several types of attack methods and demonstrates good robustness.

Although the results against adversarial attacks are promising, the paper is not well written. Especially, the notations in Section 2 are not clearly defined which baffled me a lot on how this method functions. For instance, in Equation 4, what does w_k stand for? Why there are K activation functions in a CNN, are they different? What is the meaning of a dot product between a w_k and an activation function?

Additionally, there lacks detail on how to train the transformation matrix (P in Equation 3 or T in Equation 1), and the following sentence confused me a lot: "To enforce T's positive semi-definiteness, using the eigenvalue decomposition, it can be decomposed into T'T". I understand why T needs to be PSD matrix, but how can eigenvalue decomposition decompose T into T'T? And how is this achieved during the training of a CNN? I think the authors should revise this part carefully to demonstrate the proposed methods more clearly.

---

### Official Review · AnonReviewer1 · 2018-11-08
**Interesting idea but unclear evaluation**

**Rating:** 4
**Confidence:** 4

**Review:**

The authors propose a new defense against adversarial examples based on radial basis features. Prior work has suggested that the linearity of standard convolutional networks may be a factor contributing to their vulnerability against adversarial examples, and that radial basis functions may help alleviate this weakness. The current paper builds on this idea and proposes a concrete way to add radial basis features to existing convnet architectures.

In the proposed approach, each layer of the network is augmented with a radial basis feature transform of the features in this layer. The output of this feature transform is then concatenated with the features in this layer. The centers of the radial basis features, the bandwidth, and the distance matrix are trained with the other network parameters. The distance matrix in the feature transform is used to compute the Mahalanobis distance between the features and centers in the radial basis functions.

The authors evaluate their defense experimentally on the standard MNIST dataset and two medical image datasets: X-chest14 for classification and a 2D RGB skin lesion dataset from the 2017 IEEE ISBI International Skin Imaging Collaboration (ISIC) Challenge for image segmentation. The experiments show that their method improves over an undefended network on MNIST. On X-chest14, their method improves over features squeezing (input quantization) and Gaussian data augmentation. On the image segmentation dataset, the method improves over these baselines as well as adversarial training.

While I find the overall idea interesting, I have some doubts about the experimental evaluation. For instance, the authors do not compare their MNIST numbers to the robust optimization results reported in Madry et al. (cited in the paper). Robust optimization achieves higher adversarial accuracy than the numbers reported in Table 1.

More importantly, it is unclear to what extent unmodified first-order methods are effective for the proposed defense. While the authors investigate whether their networks exhibit gradient masking / obfuscation, the left plot in Figure 3 still leaves some questions. Based on the curves for FGSM and BIM, the proposed defense would still achieve a high accuracy even against attacks with eps = 0.5. However, this would be a clear failure of the first order attacks (but not a sign of true robustness) because an adversary with eps = 0.5 can trivially defend any network by setting every input pixel to 0.5. Hence the authors should investigate what happens in the regime between eps = 0.4 and eps = 0.5.

While I support the use of non-standard datasets for evaluation, it would still strengthen the paper if the author also reported accuracy numbers on CIFAR-10. The X-chest14 and the segmentation dataset have not been frequently used in the adversarial robustness literature to the best of my knowledge. Hence it is less clear how well the proposed methods perform on these datasets.

While I find the overall idea interesting, with the current experimental evaluation I unfortunately cannot recommend accepting the paper.


Further comments:

- The distinction between "data-level" and "algorithmic-level" approaches in the introduction is unclear to me. Adversarial training can also be seen as robust optimization, which is arguably an algorithmic approach.

- At the beginning of Section 2, it would be helpful if the authors first introduced the meaning of the variables n, m, and k before using them. In general, it would be helpful if the authors described in more detail how the radial basis features are incorporated into the network.

- How is the adversarial training baseline in Section 4.2 implemented? The choice of adversary in adversarial training / robust optimization can be crucial for the robustness of the resulting model.

- Since the authors refer to the linearity of existing model as a potential weakness: there are also alternative explanations, e.g., see https://arxiv.org/abs/1801.02774 and https://arxiv.org/abs/1804.11285 .

- The test sets used in the evaluation are fairly small (150 and 200 data points). In this regime, 95% confidence intervals can be as large as +/- 8%. Hence I would recommend increasing the size of the test sets to at least 1,000.

---

### Meta-Review · Area_Chair1 · 2018-12-13
**Area chair decision**

**Confidence:** 5
**Recommendation:** Reject

**Metareview:**

Strengths of the paper:

Based on previous work suggesting that radial basis features can help defend against adversarial attacks, the paper proposes a concrete method for incorporating them in deep networks.  The paper evaluates the method on multiple datasets, including MNIST and  ISBI International Skin Imaging Collaboration (ISIC) Challenge.

Weaknesses:

Reviewers 2 and 3 felt that the paper was not clearly written, and cited several concrete questions about the method that could not be understood from the paper.  There were additional concerns of lacking comparison to existing methods, and Reviewer 1 pointed out that a competing method gave higher performance, although this was not reported in the present submission.

Points of contention:

The authors did not provide a response to the reviewer concerns.

Consensus:

All reviewers recommended that the paper be rejected, and the authors did not provide a rebuttal.